# Changes in Chemical Compositions and Antioxidant Activities from Fresh to Fermented Red Mountain-Cultivated Ginseng

**DOI:** 10.3390/molecules27144550

**Published:** 2022-07-17

**Authors:** Hee Yul Lee, Jin Hwan Lee, Eui-Cheol Shin, Du Yong Cho, Jea Gack Jung, Min Ju Kim, Jong Bin Jeong, Dawon Kang, Sang Soo Kang, Kye Man Cho

**Affiliations:** 1Department of GreenBio Science and Agri-Food Bio Convergence Institute, Gyeongsang National University, Jinju 52725, Korea; wjdald99@nate.com (H.Y.L.); eshin@gnu.ac.kr (E.-C.S.); endyd6098@naver.com (D.Y.C.); jjbbkk5612@hanmail.net (J.G.J.); minju4492@naver.com (M.J.K.); love_no_ri@naver.com (J.B.J.); 2Department of Life Resource Industry, Dong-A University, 37, Nakdong-daero 550 beon-gil, Saha-gu, Busan 49315, Korea; schem72@dau.ac.kr; 3Department of Physiology and Convergence Medical Science, Institute of Health Sciences, College of Medicine, Gyeongsang National University, Jinju 52727, Korea; dawon@gnu.ac.kr; 4Department of Anatomy and Convergence Medical Science, Institute of Health Sciences, College of Medicine, Gyeongsang National University, Jinju 52727, Korea; kangss@gnu.ac.kr

**Keywords:** mountain-cultivated ginseng, aging, lactic acid fermentation, ginsenosides, volatile flavor compounds, antioxidant

## Abstract

This study investigated changes in nutrients (fatty acids, amino acids, and minerals), ginsenosides, and volatile flavors, and antioxidant activities during food processing of mountain-cultivated ginseng (MCG) with the cocktail lactic acid bacteria. Fatty acid content increased, but the free amino acid content decreased, and minerals were practically unaffected during processing. Total phenolic and flavonoid contents and maillard reaction products increased markedly according to processing stage. The total ginsenosides levels increased from 31.25 mg/g (DMCG) to 32.36 mg/g (red MCG, RMCG) and then decreased (27.27 mg/g, at fermented RMCG) during processing. Particularly, the contents of F2 (0.31 → 1.02 → 2.27 mg/g), Rg3 (0.36 → 0.77 → 1.93 mg/g), and compound K (0.5 → 1.68 → 4.13 mg/g) of ginsenosides and β-panasinsene (17.28 → 22.69 → 31.61%), biocycloelemene (0.11 → 0.84 → 0.92%), δ-cadinene (0.39 → 0.5 → 0.94%), and alloaromadendrene (1.64 → 1.39 → 2.6%) of volatile flavor compounds increased during processing, along with to the antioxidant effects (such as DPPH, ABTS, and hydroxyl radical scavenging activities, and FRAP). This study may provide several choices for the use of ginseng in functional foods and functional cosmetics.

## 1. Introduction

Ginseng (*Panax ginseng* C.A. *Meyer*) has been long used worldwide as a dietary supplement and/or health functional food in traditional herbal medicine. Numerous researches have also shown that this plant plays an important role in human health with beneficial properties such as antidiabetic, anticancer, anti-aging, and anti-inflammatory [1,2,3]. There are three kinds of ginseng according to cultivation methods and environmental conditions: cultivated ginseng (CG), mountain-cultivated ginseng (MCG), and mountain wild ginseng (MWG). MCG proliferates under natural conditions in the forest and demands a long growth period (>3 years) [1,2,3,4]. In general, ginseng is consumed either fresh, or as red, black, and fermented red ginseng in food and medical industries because of its many metabolite constituents [5,6,7,8]. Ginseng saponins, named ginsenosides, are responsible for their pharmacological and biological effects, including antioxidant, anti-inflammatory, anticancer, and immune activities [9]. Among various compositions, ginsenosides Rb1, Rb2, Rd, Re, and Rg1 are major ones, accounting for approximately 80% of ginsenosides, and result from diverse processing methods such as, including steaming, aging, and fermentation for the conversion of pharmacologically and biological active ginsenosides [1]. However, they have low pharmacological and biological activities as well as low absorption of the human gut [10]. On the other hand, ginsenosides Rg3, Rh2, F2, and compound K (CK) are minor ginsenosides, with higher pharmacological activities and which allow higher uptake than the major ginsenosides [11,12].

Volatile flavor compounds (VFCs) are one of the most important quality factors of roasted/or fermented foods and a key determinant of consumer acceptance [13]. Plant VFCs are increasingly recognized for their materials have aromatic, antifungal, insecticidal, biological, and therapeutic properties and are of great interest to the perfume, cosmetic, and food industries [14]. Even though there have been several articles on ginsenoside and VFC contents with environmental conditions, to the best of our knowledge, there has been no comprehensive information on MCG using processing techniques.

Commonly, lactic acid bacteria (LAB) are gram-positive bacteria of potential probiotics. LAB has been used for the production of fermented foods, like yogurt [15] and kimchi [16]. Previous research reported the ginsenoside conversion using a variety of LAB, such as *Lactobacillus plantarum*, *Lactobacillus brevis*, and *Bifidobacterium longum* [17,18]. Most studies investigated this conversion and the changes in pharmacological and biological activities by heat treatment/or fermentation [1,17,18,19]. However, there are few studies on the primary metabolites and VFCs and antioxidant activities according to MCG processing steps. For these above observations, our work was designed to demonstrate the primary and secondary metabolites as well as antioxidant properties under processing skills of aging and fermentation from MCG.

The main purpose of the present research study was to determine the changes to nutritional and/or functional factors, including fatty acids (FAs), free amino acids (FFAs), minerals, ginsenosides, and VFCs, during the food processing with cocktail *L. plantarum* P1201 and *L. brevis* BMK484. In addition, our study evaluated changes in physicochemical properties (such as pH, acidity, and soluble solids), total phenolic (TP), total flavonoid (TF) contents, and maillard reaction (MR) products. We also investigated the fluctuations of antioxidant effects including DPPH, ABTS, hydroxyl, and FRAP in aging and fermentation processes of MCG.

## 2. Results and Discussion

### 2.1. Change in Physicochemical Properties of Dried MCG, Red MCG, and Fermented Red MCG

The manufacturing process of dry MCG (DMCG), red MCG (RMCG), and fermented RMCG (FRMCG) is shown in Figure 1. The pH decreased during food processing from 5.4 (DMCG) to 4.44 (FRMCG), corresponding to an increase in acidity from 0.21 to 0.51. In addition, the contents of soluble solids increased a little in the RMCG sample (3.2 brix) and decreased slightly in FRMCG (Figure 2). Based on the above results, we confirmed that the environmental stresses of processing methods may be important influences on variation ratios regarding action glycoside hydrolases from DMCG [11,20]. During a typical lactic acid fermentation, the pH decreases while acidity increases [1,21,22]. This finding has also been reported during the lactic acid fermentation of ginseng and/or red ginseng [11,18,23], and was corroborated by our results.

### 2.2. Changes in Nutrients According to the Processing Steps of MCG

As shown in Table 1, the levels of 16 FAs, including 6 saturated FAs (SFAs) and 10 unsaturated FAs (USFAs), were analyzed and profiled in various MCG samples (DMCG, RMCG, and FRMCG). The total FAs, SFAs, and USFAs content increased at RMCG to 791.4, 358.1, and 433.1 mg/100 g, respectively; and decreased to 749.7, 333.9, and 415.8 mg/100 g at FRMCG step. USFAs levels were approximately 260%, 221%, and 224% higher (417.6, 433.1, and 415.8 mg/100 g, respectively) than SFAs (260.1, 358.1, and 333.9 mg/100 g), respectively. In particular, the contents of linoleic acid were the highest (333.1, 337.6, and 329.1 mg/100 g) follow by palmitic acid (166.1, 215.2, and 208.7 mg/100 g) > stearic acid (58.6, 101.4, and 90.7 mg/100 g) > oleic acid (35.8, 44.9, and 34.0 mg/100 g) > α-linoleic acid (28.3, 29.7, and 30.7 mg/100 g). Liu et al. [24] reported that MCG appeared to have a higher FA content than in CG/or MWG. Of the 19 FAs in ginseng root, linoleic acid is the most predominant in three *Panax species*, including *P. ginseng*, *P. notoginseng*, and *P. quinquefolus*, accounting for 43–53% of total FAs [25]. In the present research, the USFAs were approximately 2.6-folds higher than SFAs in DMCG roots: similar to previous CG results [26]. However, the proportion of USFAs with respect to SFAs detected during food processing steps increased. A previous study reported that fermentation strain did not affect total FAs [27]. Our results slightly differ from previous data in that the total FA content of ginseng seeds and MCGS did not change according to the fermentation process [1,27]. The total FA contents increased after fermentation in this study and their differences may be considered as the different kinds of ginseng and microorganisms as well as fermentation environmental conditions.

Thirty free amino acids (FAAs), 22 non-essential AAs (NEAAs), and 8 essential AAs (EAAs), are shown in Table 2. Total AAs, NEAAs, and EAAs decreased markedly from 2172.68 mg/100 g (DMCG stage) to 756.3 mg/100 g (FRMCG stage) in food processing. NEAAs levels were approximately 419%, 486%, and 415% higher (1115.93, 440.74, and 351.31 mg/100 g than those of EAAs (266.02, 90.59, and 84.58 mg/100 g) in each step, respectively. At DMCG steps, arginine was the predominant amino acid, constituting approximately 52% of total FAAs, and decreasing greatly from 1115.93 mg/100 g to 351.31 mg/100 g over-processing. In addition, the aminobutyric acid (GABA) content sharply decreased from 188.99 mg/100 g to 9.5 mg/100 g at the RMCG step increasing markedly to 100.61 mg/100 g at FRMCG.

Some researchers have reported reduced the total AAs content after steaming, aging, and fermenting when compared to raw ginseng [8,28,29]. Wan et al. [8] ranked total AAs on decreasing order as follows: fresh ginseng > frozen ginseng > white ginseng > stoved ginseng > red ginseng > black ginseng. Of all AAs, arginine, and GABA (non-protein amino acid) were abundant in the raw and processed ginseng. In particular, arginine and GABA have been identified as the most abundant essential amino acid in various processed ginseng products [8,28]. Recently, Lee et al. [1] reported arginine (33%) and GABA (18%), as the most common AAs of MCG sprout. Moreover, these values exhibited a significant decrease of 187.7 → 164.9 → 74.9 mg/100 g and 100.3 → 35.7 → 13.5 mg/100 g in aging and fermentation steps. Furthermore, considering that arginine is a promoter of human cell division and GABA an inhibitory neurotransmitter in the central nervous system, they can be considered important quality indicators of ginseng products [8,29]. The fluctuations of individual and total AAs may be also associated with various parameters concerned with the process techniques [30]. The amounts of minerals barely changed over food processing steps. The major mineral observed was potassium (K) with 16.07, 15.97, and 15.93 mg/100 g, followed by phosphorus (P) > calcium (Ca) > magnesium (Mg) > sulfur (S). The minor minerals were followed by sodium (Na) > aluminium (Al) > iron (Fe) > zinc (Zn) > manganese (Mn) > copper (Cu) (Table 3).

The main minerals in ginseng are P, Ca, K, and Mg [31] and Kim et al. [32] reported slightly lower total mineral contents in red ginseng compared to fresh ginseng, similar to our results. The present data are also in agreement with previous research reporting Ca, K, P, and Mg as main minerals in ginseng [33]. Furthermore, our results were in agreement with previously reported research that the total minerals of red ginseng decreased slightly than fresh ginseng [34]. In particular, we believe that the increased rates of minerals in the FRMCG sample may be related to the degradation and transformation of other nutritional compositions through microbial growth during the fermentation process [9,10,31].

### 2.3. Change in TP and TF Contents and MR Products during the Food Processing Steps of MCG

The TP and TF contents and MR products greatly increased during the processing methods of MCG with the cocktail LAB (Figure 3). First, the TP contents increased dramatically from the DMCG of 2.33 to 5.61 GAE mg/g till the FRMCG. Second, the levels of TF were detected the 0.17, 0.66, and 0.86 RE mg/g at DMCG, RMCG, and FRMCG samples. Finally, the levels of MR product also increased sharply from 1.29 (OD_420 nm_) at DMCG step to 2.29 (OD_420 nm_) at FRMCG step. Recent literature has demonstrated that the TP contents of ginseng increased in the lactic acid fermentation using *L. plantarum* KCCM 11613P [11].

Previously, higher TP contents of fermented black ginseng with *S. cerevisiae* than raw and/or black ginseng were reported [33]. These phenomena may be positively correlated with changes in acid or estrolytic enzymes through the release of insoluble ester bounds (including bound phenolic and/or flavonoids) increasing TP and TF contents [11,34]. Additionally, the levels of MR products increased by thermal and fermentative processing; thus, the decrease in total AAs is caused by the extent of MR. Lee et al. [1] recently published that the content of TP and TF and MR products increased during MCG sprout processing steps. Our findings are consistent with the data of the previous studies [1,30]. In particular, numerous studies have reported a positive correlation between potential antioxidant activities with the TP and TF contents and MR products [1,22,35]. Therefore, we suggest that these elements including TP and TF contents and MR products have radical scavenging activities and reducing power.

### 2.4. Change in Ginsenoside Compounds According to the Food Processing Steps of MCG

Ginsenosides are typically classified based on chemical structures as follows: the four-ring dammarne type and five-ring oleanane type. In addition, dammarne types are divided into PPD and PPT forms [36]. Previously, the following PPD bioconversion according to the fermentation process in ginseng/or red ginseng was reported as Rb1 → Rd → F2 → Rh2/or Rb → Rd → Rg3 [23], Rb1 was directly converted to Rg3 [37], and Rb1 conversion to compound K (Rb1 → Rd → F2 → compound K) [38]. The typical chromatograms of 21 ginsenoside peaks gained, which appeared significant differences in different food processing steps (Figure 4), and their contents are shown in Table 4. The total ginsenoside contents through the processing methods increased slightly from 31.25 mg/g (at DMCG step) to 32.36 mg/g (at RMCG step), and then decreased (27.47 mg/g, at FRMCG step). The protopanaxatriol (PPT) types were barely affected from DMCG to RMCG (8.79 mg/g to 8.2 mg/g) at the middle step and decreased markedly thereafter (5.32 mg/g, at the finial FRMCG step). Among PPT types, the amounts of ginsenoside Rg1, Re, and Rf decreased sharply with 2.16 → 1.64 → 0.52, 4.4 → 3.25 → 0.95, and 0.79 → 0.73 → 0.62 mg/g between aging and fermentation, while the ginsenoside Rg2, Rh1, and PPT contents increased with 0.48 → 0.73 → 1.01, 0.96 → 1.19 → 1.41, and nd → 0.64 → 0.81 mg/g, respectively. On the other hand, protopanaxadiol (PPD) types were almost unchanged in aging and fermentation processes with 20.24 (DMCG) → 25.32 (RMCGS) → 25.76 mg/g (FRMCG). Among them, the values of ginsenosides Rb1 (8.7 → 9.01 → 4.69 mg/g), Rc (4.2 → 4.31 → 2.05 mg/g), and Rh2 (1.62 → 1.47 → 0.78 mg/g) decreased greatly, whereas Rd2 (1.81 → 2.35 → 2.77 mg/g), F2 (0.31 → 1.02 → 2.27), Rg3 (0.36 → 0.77 → 1.93 mg/g), CK (0.5 → 1.68 → 4.13), and Rh2 (nd → 0.26 → 0.5 mg/g) increased in these above processes. Meanwhile, ginsenoside Rd increased at RMCG (2.35 mg/g), and decreased at FRMCG (2.77 mg/g), but the content of PPD decreased the 0.31 mg/g at RMCG, after this increased the 0.5 mg/g at FRMCG stage. Finally, the amount of ginsenoside Ro (oleanane type) decreased dramatically from 2.22 mg/g (at DMCG) to 0.77 mg/g (RMCG) and increased thereafter (1.09 mg/g, at FRMCG).

Consequently, we found intermediate metabolites of Rd and F2 before initial metabolites of Rb1, Rb2, and Rc were converted to final metabolites of Rg3, compound K, and Rh2, indicating the following conversion steps of Rb1/or Rb2/or Rc → Rd → Rg3 → Rh2 or Rb1/or Rb2/or Rc → Rd → F2 → compound K/or Rh2 at FRMCG (Figure 5A). For PPT type, there were intermediate ginsenosides of Rg1, Rg2, Rh1, and F1 generated before ginsenosides Re, F3, and F3 were transferred to aglycone PPT, indicating the following the transformation steps of Re → Rg1/or Rg2/or Rf → Rh1/or F1 → PPT (aglycone form) or F5/or F3 → F1 → PPT in FRMCG step (Figure 5B). Several studies considered it due to hydrolytic enzymes (such as β-glycosidase etc.) and metabolite products (such as lactic acid etc.) of LAB, like *Enterococcus* spp., *Lactobacillus* spp., *Leuconosotoc* spp., *Pediococcus* spp. and *Bifidobacterium* spp., in ginseng/or red ginseng [17,18,19]. Furthermore, our results agree with recent showing that the ginsenosides of Rd2, F2, Rg3, and CK fermented red MCG sprouts increased sharply than fresh MCG sprouts [1].

### 2.5. Change in VFCs According to the Processing Steps of MCG

Little is known about VFCs in MCG during aging and fermentation processes. During food processing, the typical gas chromatograph-mass spectrometer (GC-MS) chromatograms concerned with the change in 39 VFCs are shown in Table 5. A total of 28 VFCs were detected in DMCG and the number of VFCs decreased to 22 (RMCG step) and 21 (FRMCG step) thereafter. Particularly, β-panasinsene appeared predominant with 17.28% (MCG step), 22.69% (RMCG step), and 31.61% (FRMCG step) followed by β-elemene (13.6% → 8.82% → 13.7%), calarene (12.9 → 9.54 → nd), carypohyllene (12.9 → 8.5 → 13.82%), bicyclogermacrene (11.97 → 4.9 → 4.8%), and selina-4,11-diene (9.5% → nd → nd). The eight compounds, including bicycolene, drina-8,12-9,11-diene, 4-1-methyl ethyl benzaldehyde, α-gurjunene, alloaromadendrene, β-selinene, germacrene, and δ-cadinene, were commonly detected as minor VFCs at different three steps (Table 5).

These phenomena suggested that variations of VFCs in MCG samples may be connected with the degradation of amino acids under aging and fermentation processes [39]. In the previous research, Cho et al. [40] previously reported that the main VFCs were detected in bicyclogermacrene, (E)-β-farnesene, β-panasinsene, calarene, α-humulene, and β-elemene in *P. ginseng*. A total 47 VFCs among which, the 3-acetyl-1-(3,4-dimethoxyphenyl)-5-ethyl-4,5-dihydro-7,8-dimethoxy-4-methylene-3H-2,3-benzodiazepine, 2-furanmethanol, and 1,2-benzenedicarboxylic acid dibutyl ester in fresh, white, and red ginseng, respectively [14]. A previous researcher found that the environmental conditions of processing methods may markedly influence VFC profiles and their concentrations [41,42], and that VFCs decrease may relate to amino acid degradation and glucose conversion during processing periods [39]. Therefore, the VFCs of MCGs decreased considerably with processing systems and their contents may be not an excellent factor in the quality of processed MCGs. This work was the first to demonstrate the degree of interrelation between the processing method and the VFC profile.

### 2.6. Change in Antioxidant Effects According to the Processing Steps of MCG

As shown in Figure 6, we assayed antioxidant activities in 50% ethanol (EtOH) extracts at different processing steps (DMCG, RMCG, and FRMCG) and compared the DPPH, ABTS, and hydroxyl radical scavenging activities and FRAP assay. Both RMCG and FRMCG steps had greater antioxidant effects than DMCG stage, particularly DPPH radical scavenging activity when treated at the same concentration. The effect of DPPH radical scavenging activity of 50% EtOH extracts at 1 mg/mL was in the following: FRMCG step (70.2%) > RMCG step (67.6%) > DMCG step (53.4%). The scavenging activity of ABTS radical in 50% EtOH extracts of 0.5 mg/mL in the order of FRMCG (78.4%), RMCG (74.8%), and DMCG (28.4%). The hydroxyl radical scavenging activity in the 50% EtOH extracts at 1 mg/mL followed the order of processing steps DMCG (24.1%), RMCG (51.9%), and FRMCG (57.3%). The FRAP assay in 50% EtOH extracts at 1 mg/mL showed the following trend: FRMCG step (1.2) > RMCG step (1.1) > DMCG step (0.5) (Figure 6). Based on the above data, the antioxidant properties of MCG showed significant differences with the increased ratios during aging and fermentation processes. In previous reports, the antioxidant effects of edible and medicinal plants (including soybean, ginseng, MCG sprouts, etc.) increased with thermal and fermentative processes [1,11,22]. Fermented ginseng seeds with *Bacillus subtilis*, *Pediococcus pentosaceus*, and *Lactobacillius gasseri* had higher antioxidant activity than unfermented ginseng seeds [9]. Jung et al. [33] reported that the antioxidant activity was dramatically increased when black ginseng was fermented by *Saccharomyces cerevisiae* than the control (unfermentation) black ginseng. Moreover, fermentation of red ginseng extract with *L. plantarum* produced stronger antioxidant activity than in the original red ginseng extract [11]. Lee et al. [1] recently announced that the antioxidant activities (such as DPPH, ABTS, and hydroxyl radical scavenging activities) of FAMCG sprout increased by approximately 20% more than MCG sprouts with the rank order of ABTS > DPPH > hydroxyl.

## 3. Materials and Methods

### 3.1. Plant, Microorganisms, Medium, Chemicals, and Instruments

The MCG (more than five years) was harvested and supplied in 2018 from GinsengBio Association Co. (Hamyang-gun, Gyeongsangnam province, Korea). For the fermentation of starters, including *L**. plantarum* P1201 and *L. brevis* BMK484, were isolated from fermented beverage plant extracts and *mullkimchi* by previously reported methods, respectively [20]. They were pre-cultured in lactobacilli MRS broth or agar (MRSB or MRSA, Difco, Becton Dickinson Co., Sparks, MD, USA). The 21 ginsenoside standards were obtained from KOC Biotech Co., Ltd. (Daejeon, Korea). The antioxidant reagents and high press liquid chromatography (HPLC) solvents were purchased from Sigma-Aldrich Chemical Co. (St. Louis, MO, USA) as described previously in Hwang et al. and Lee et al. [15,22].

The pH and acidity values were measured using a pH meter (Model 3510, Jenway, UK). TP and TF contents, MR products, and antioxidant assay were analyzed using UV-Vis. absorption spectra on a Shimadzu Scientific Korea Corp. (UV-1800 240V, Seoul, Korea). The FAs content was analyzed using a GC 7980 system (Agilent Technologies Inc., Wilmington, DE, USA) containing a flame ion detector (FID) and SP-2560 capillary column (100 m × 0.25 mm, 0.25-μm film thickness; Sigma-Aldrich Co., St. Louis, MO, USA). The analysis of FAAs and minerals was measured using an automatic amino acid analyzer (Hitachi High-Technologies Corp., Tokyo, Japan) and an inductively coupled plasma spectrometer (ICP, NexION 350 ICP MS, PerkinElmer Inc., Waltham, MA, USA) respectively. The analysis of ginsenoside was quantified using an HPLC 1260 system (Agilent Technologies Inc., Waldbronn, Germany) including the Agilent 1260 diode-array detector, quaternary pump, autosampler, and TSK-ODS100Z (Tosoh Corp., Tokyo, Japan). The VFCs were analyzed using a gas chromatograph-mass spectrometer (GC-MS, GC-7890A, MSD-5975C, Agilent Co., Waldbronn, Germany) containing an FID and HP-5MS (30 m × 0.25 mm, 0.25 µm film thickness).

### 3.2. Preparation of Dry MCG, Red MCG, and Fermented Red MCG

The MCG was washed three times and then steamed for 60 min at 95 ± 2 °C. The steamed MCG was placed in an aging container and aged at 75 ± 2 °C for 72 h. This process was repeated three times. The red MCG (RMCG) was placed in a dry oven at 50 ± 2 °C for 72 h to evaporate the water and then was crushed by a grinder to make RMCG powder. The powdered RMCG (100 g) was transferred separately into a 500 mL stainless container containing 200 mL tap water with 2% sucrose. The mixture was sterilized in an autoclave at 121 °C for 15 min and after was cooled at 35 ± 2 °C. The pre-cultured *L. plantarum* P1201 and *L. brevis* BMK484 were inoculated at the cell density of 8.6 log cfu/mL and 8.2 log cfu/mL, respectively; after the RMCGs were fermented at 35 ± 1 °C for 120 h (fermented RMCG, FRMCG) (Figure 1). The dry MCG (DMCG), RMCG, and FRMCG samples were freeze-dried and stored at -70 ± 1 °C until analysis.

### 3.3. Determination of Physicochemical Properties and Viable Cell Numbers

Physicochemical properties (pH and acidity) and viable cell numbers were previously determined according to Hua et al. and Lee et al. [5,21]. Briefly, the pH was performed on the pH meter and the acidity was measured upon titration with a 0.1 N NaOH solution and converted to lactic acid values. Viable cell numbers were determined by the method of ten series dilution. Namely, one gram of each sample was dissolved in nine milliliters of sterilized distilled water and the diluted suspension was spread on MRSA plates. The plate was incubated at 35 ± 1 °C for 48 h; then, the colonies were measured on MRSA plates.

### 3.4. Analysis of FAs

FAs were analyzed according to previously reported data [22,25]. Shortly, 2 mL of different samples, including DMCG, RMCG, and FRMCG, were mixed with 3 mL of 0.5 N NaOH in methanol (MeOH) and heated at 100 °C for 10 min. After cooling to 35 ± 2 °C, the 2 mL of 14% borontrifluride in MeOH (BF_3_/MeOH) were added and the mixture vortexed and heated at 100 °C for 30 min for FA methylation. Next, the 5 mL of saturated NaOH solution (28%, *w*/*v*) and 2 mL of isocatone were added and mixed; then, the supernatant was collected. The FA contents of supernatant were measured using a GC containing mobile phase with nitrogen gas. The gas was flowed 1 mL per minute with injecting sample (20 µL). The initial column oven temperature was set at 140 °C and maintained for 5 min before ramping up at 4 °C/min to 230 °C where it was maintained for 35 min.

### 3.5. Analysis of FAAs

The FAAs were according to Kuo et al. and Wan et al. [8,30]. The three samples (1 g) were added to 4 mL of distilled water and after heating on a heating block for induced hydrolysis at 60 ± 2 °C for 2 h. Next, 1 mL of a 10% 5-sulfosalicylic acid was added, and the samples were vortexed for 1 min and cooled at 4 ± 1 °C for 2 h. To collect supernatants, the mixtures were centrifuged at 3000× *g* for 30 min and concentrated using a rotary vacuum evaporator at 60 ± 2 °C. Finally, the concentrates were dissolved in lithium buffer (pH 2.2) and filtered. FAA contents were analyzed using an amino acid automatic amino acid analyzer.

### 3.6. Analysis of Minerals

The analysis of minerals was performed according to Park et al. [31]. The 0.5 g of the processed samples (DMCG, RMCG, and FRMCG) was precisely weighed and the 10 mL of a 70% nitric acid solution was added and mixed. The mixture was decomposed by microwaves; then, after the degradation products were adjusted to 50 mL using tertiary distilled water. This sample was filtered and analyzed by ICP.

### 3.7. Analysis of TP and TF Contents and MR Products

The TP and TF contents were previously performed according to Lee et al. [21]. The 1 g of each sample (such as DMCG, RMCG, and FRMCG powders) was added to 20 mL of the 50% ethanol and the mixtures were extracted at 70 ± 2 °C for 1 h. The extract was filtered through a 0.45 µm membrane filter to recover the supernatant. To determine the TP and TF contents, the diluted solution of the 50% ethanol extracts (0.5 mL) was mixed with 0.25 mL Folin-Ciocalteu reagent (0.8 mL) and 0.5 mL of 25% Na_2_CO_3_ or the mixtures of 1.0 mL diethylene and 0.1 mL 1 N NaOH, respectively. This mixture was incubated at 30 ± 1 °C or 37 ± 1 °C for 1 h; then, the final reaction products were determined at 750 or 420 nm using a UV-Vis spectrophotometer, respectively. MR products were analyzed by non-enzymatic browning as described previously [43]. The 10 g of tertiary distilled water was added to 1 g of the sample followed by 1 h extraction. The filtered extract was measured at 420 nm.

### 3.8. Analysis of Ginsenosides

To analyze ginsenosides, 1 g of each powder, including DMCG, RMCG, and FRMCG, was dissolved into 20 mL of 50% methanol and the mixtures were extracted at 70 ± 2 °C for 1 h. The extract was filtered through a 0.45 µm membrane filter to recover the supernatant; this process was repeated two times. Next, each extract-supernatant was concentrated and dried using a rotary evaporate. Finally, dried extracts were dissolved into the 2 mL of acetonitrile:H_2_O (8:2, *v*/*v*) and dissolved. The dissolved samples were filtered through a 0.45-µm membrane filter for HPLC analysis. The analysis of ginsenoside derivates was modified and performed as previously described in Shibata et al. [44] and Jung et al. [11]. The flow rate, injection volume, column oven temperature, were adjusted at 1 mL/min, 20 µL, UV 203 nm, and 40 °C, respectively. The mobile phase was composed of solvent A (water) and solvent B (acetonitrile) using the following gradient program: 0 min 19% B, 10 min 19% B, 15 min 20% B, 40 min 23% B, 42 min 30% B, 75 min 35% B, 80 min 70% B, 90 min 90% B, and 100 min 90% B.

### 3.9. Analysis of VFCs

VFCs in the three samples were extracted and analyzed using solid phase-micro extraction with a Headspace (Autosampler, HS-7697A, Agilent Technologies, Santa Clara, CA, USA) according to El-Aty et al. [14] and Cho et al. [40] with slight modification. Exactly, 5 mL of the samples were sealed in a 20 mL glass vial with an aluminum cap. Then, 100 μm polydimethylsiloxane 100 rpm in fiber, the samples were adsorbed at 100 °C for 8 min. The sample was injected for 0.2 min using fractional injection (10:1) and helium (He) was used as carrier gas with flowing at 1 mL/min. The initial column oven temperature was set at 40 °C and maintained for 3 min before ramping up at 5 °C/min to 280 °C and was held for 100 min. The temperature of the injector and quadrapole was 110 °C and MS was performed in scene mode (range 50 to 550) using electron impact ionization (69.9 eV). The MS of each peak was confirmed from the obtained chromatogram and each VFC was confirmed by comparing it with the GC-MS NIST library.

### 3.10. Determination of Antioxidant Effects

The 10 g of each powder, such as DMCG, RMCG, and FRMCG, was dissolved into 200 mL of 50% EtOH and the mixtures were extracted at 40 ± 2 °C for 5 h. The extract was filtered through a 0.45 µm membrane filter to recover the supernatant; this process was repeated two times. To prepare the extract-powders, each of extract-supernatant was concentrated using a rotary evaporate and then dried by a freeze dryer. Finally, the extract-powder was diluted to 0.25, 0.5, and 1.0 mg/mL in 50% EtOH and filtered.

The antioxidant effects against DPPH, ABTS, and hydroxyl (•OH) radical scavenging activities were measured according to Cho et al. [28] and Lee et al. [9]. For the DPPH and ABTS assays, the diluted solution included 50% EtOH extract-concentrates mixed with the DPPH reaction solution, ABTS reaction solution, or hydroxyl reaction solution, respectively. The final reaction products were determined at 525 nm, 730 nm, or 510 nm using a UV-Vis spectrophotometer, respectively.

The FRAP assay was previously performed as in Nooshkam et al. [43] and Lee et al. [1]. The FRAP solution was prepared by mixing 300 mM acetate buffers (pH 3.6), 10 mM TPTZ in 40 mM HCl, and 20 mM FeCl_3_ at 10:1:1 (*v*/*v*/*v*). The prepared solution was preliminarily incubated at 37 °C for 15 min; then, 0.05 mL diluted samples and 0.95 mL FRAP solution were mixed and incubated at 37 °C for 15 min. The sample was measured at 593 nm.

### 3.11. Statistical Analysis

The nutrient compounds and antioxidant activities were expressed as the mean ± SD (standard derivation) of five measurements. The significant differences among samples were determined by Tukey’s multiple test (*p* < 0.05) using the Statistical Analysis System (SAS) software (ver. 9.4; SAS Institute, Cary, NC, USA).

## 4. Conclusions

Our work demonstrated for the first time that an important information concern to changes in primary (FFAs, FAs, and minerals) and secondary (ginsenosides and VFCs) metabolites as well as antioxidant properties of processed MCG by aging and fermentation. In physicochemical factors under processing steps, the pH values decreased from 5.4 (DMCG step) to 4.44 (FRMCG step) during processing, while the acidity increased from 0.21% to 20.51%. During food processing, the total FA contents (677.7 → 749.7 mg/100 g) increased from 0.2% to 20.5%. The total FAA contents decreased from 2172.68 to 756.38 mg/100 g. The values of total ginsenosides (31.25 → 27.47 mg/g) and PPT type (8.79 → 5.32 mg/g) decreased slightly, but the PPD type (20.64 → 21.06 mg/g) was hardly affected. In particular, the bioconversion of PPD and PPT by lactic acid fermentation in AMCGS was shown to follow the conversion Rb1/or Rb2/or Rc → Rd → Rg3 → Rh2 or Rb1/or Rb2/or Rc → Rd → F2 → compound K/or Rh2 (PPD type) and Re → Rg1/or Rg2/or Rf → Rh1 → PPT (aglycone form) in FAMCG stage. Additionally, the amounts of Rg3 (0.36 → 1.93 mg/g, 5.4-folds) and CK (0.5 → 4.13 mg/g, 8.3 times) markedly increased. In the VFC contents, β-panasinsene (17.28 → 31.61%, 1.8-folds), biocycloelemene (0.11 → 0.92%, 8.4-folds), δ-cadinene (0.39 → 0.94%, 2.4-folds), and alloaromadendrene (1.64 → 2.6%, 1.6-folds) exhibited increase rates with high variations by comparing the other compositions. The TP (2.3 → 5.61 mg/g) and TF (0.17 → 0.86 mg/g) contents and MR products (1.292 → 2.295 OD_420 nm_) increased sharply over processing steps. Moreover, the antioxidant capacities of processed FRMCG increased by about 20% more than MCG with the rank order as follows: DPPH > ABTS > hydroxyl > FRAP. In summary, processed DMCG samples may be considered as potential ginseng nutritional agents for future commercial applications. Also, we believed that processed DMCG sources can be key materials in the ginseng quality for the development of human functional foods. Future research is needed to document the beneficial human health biological activities in processed DMCG for the advancement of food and medical industrious.

## Figures and Tables

**Figure 1 molecules-27-04550-f001:**
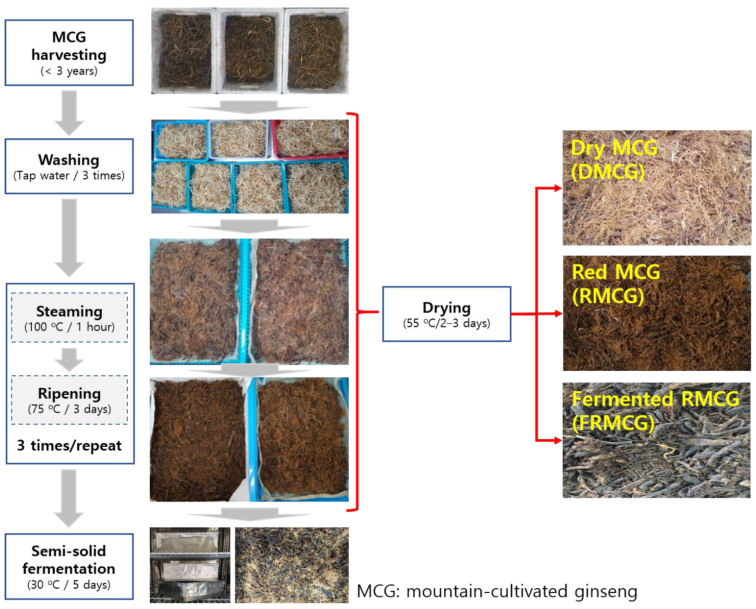
Photograph of food processing steps of mountain-cultivated ginseng. DMCG, dry mountain-cultivated ginseng; RMCG, red mountain-cultivated ginseng; and FRMCG, fermented red mountain-cultivated ginseng.

**Figure 2 molecules-27-04550-f002:**
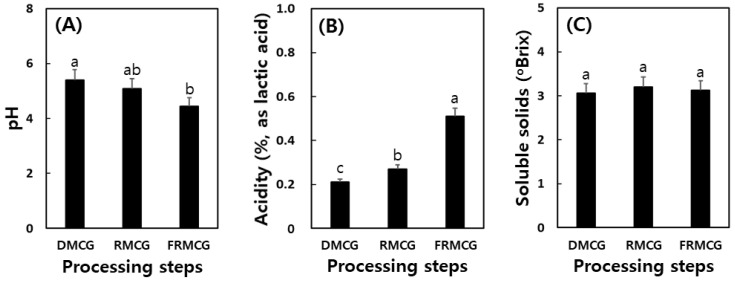
Change of pH (**A**), acidity (**B**), and soluble solids (**C**) during food processing steps of mountain-cultivated ginseng by the cocktail lactic acid bacteria. DMCG, dry mountain-cultivated ginseng; RMCG, red mountain-cultivated ginseng; and FRMCG, fermented red mountain-cultivated ginseng. All values are present as the mean ± SD of pentaplicate determination. Different letters correspond to the significant differences relating to the processing steps using Tukey’s multiple tests (*p* < 0.05).

**Figure 3 molecules-27-04550-f003:**
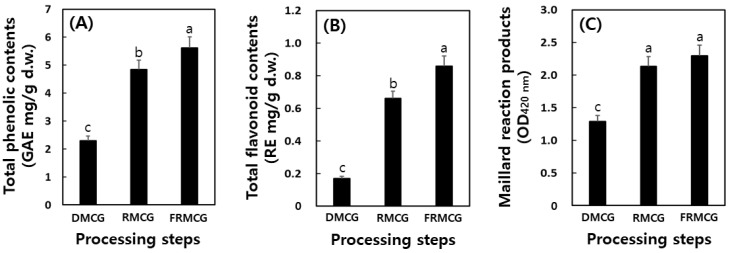
Change of the total phenolic (**A**) and total flavonoid (**B**) contents and maillard reaction products (**C**) during food processing steps of mountain-cultivated ginseng by the cocktail lactic acid bacteria. DMCG, dry mountain-cultivated ginseng; RMCG, red mountain-cultivated ginseng; and FRMCG, fermented red mountain-cultivated ginseng. All values are present as the mean ± SD of pentaplicate determination. Different letters correspond to the significant differences relating to the processing steps using Tukey’s multiple test (*p* < 0.05).

**Figure 4 molecules-27-04550-f004:**
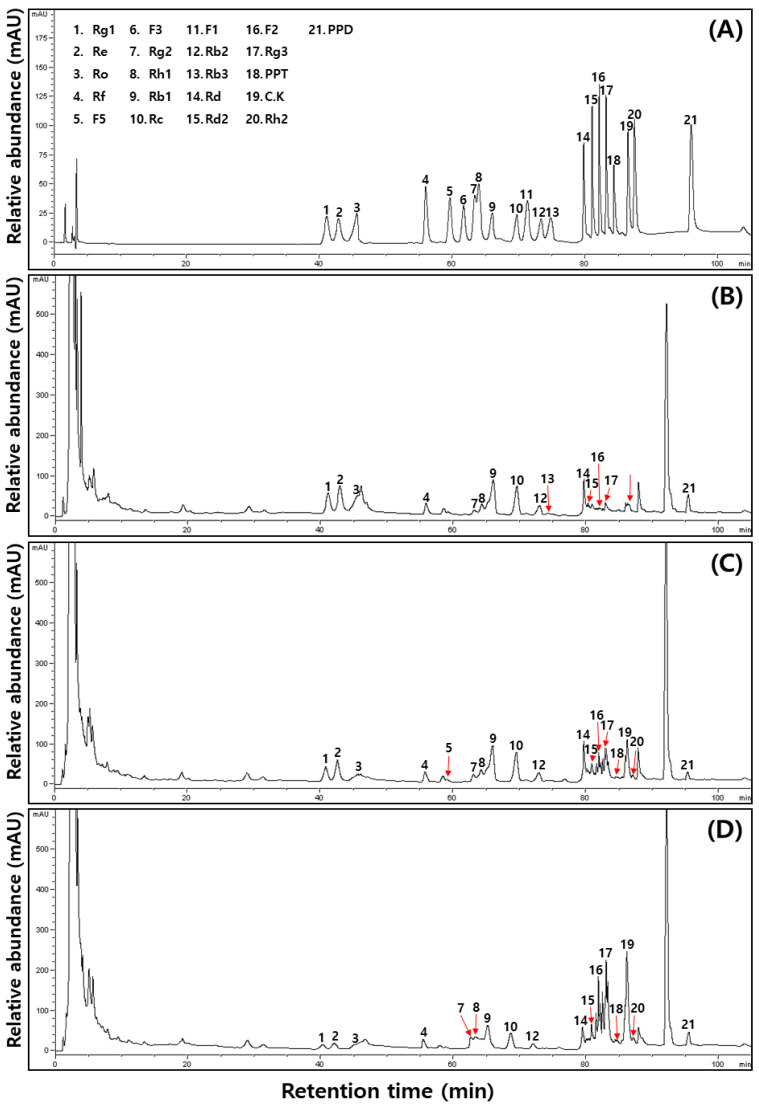
Typical ginsenosides HPLC chromatogram of the 50% ethanol extracts during food processing steps of mountain-cultivated ginseng by the cocktail lactic acid bacteria. (**A**) Standard; (**B**) 50% ethanol extract of dry mountain-cultivated ginseng (DMCG); (**C**) 50% ethanol extract of red mountain-cultivated ginseng (RMCG); and (**D**) 50% ethanol extract of fermented red mountain-cultivated ginseng (FRMCG).

**Figure 5 molecules-27-04550-f005:**
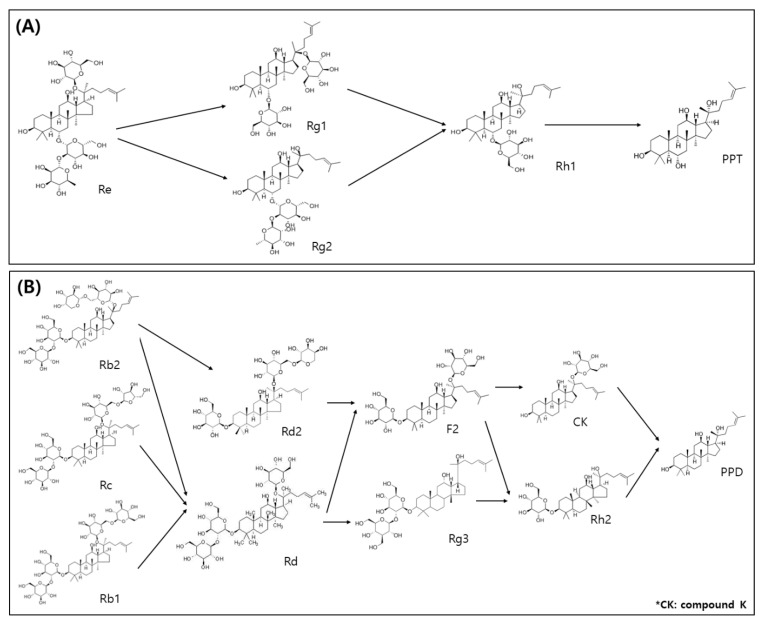
Bioconversion pathway of ginsenoside compositions in mountain-cultivated ginseng sprouts during the processing steps. (**A**) Pathway of protopanaxatriol (PPT) type; and (**B**) Pathway of protopanaxadiol (PPD) type.

**Figure 6 molecules-27-04550-f006:**
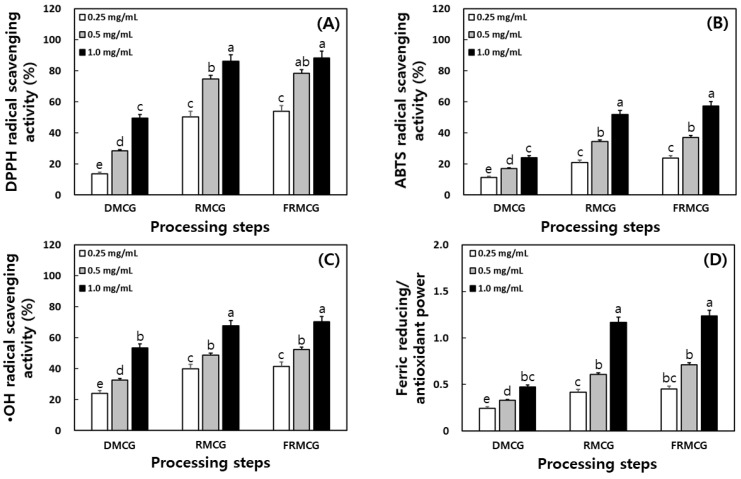
Comparison of antioxidant properties in the 50% ethanol extracts through processing steps of mountain-cultivated ginseng sprouts. (**A**) DPPH radical scavenging activity; (**B**) ABTS radical scavenging activity; (**C**) hydroxyl radical scavenging activity; and (**D**) ferric reducing/antioxidant power. All values are present as the mean ± SD of pentaplicate determination. Different letters correspond to the significant differences relating to the processing steps using Tukey’s multiple test (*p* < 0.05). DMCG, dry mountain-cultivated ginseng; RMCG, red mountain-cultivated ginseng; and FRMCG, fermented red mountain-cultivated ginseng.

**Table 1 molecules-27-04550-t001:** Change of the fatty acid contents during food processing steps of mountain-cultivated ginseng with the cocktail lactic acid bacteria.

Contents a (mg/100 g d.w.)	Processing Steps b		
	DMCG	RMCG	FRMCG
Saturated fatty acids			
Myristic acid (C14:0)	1.2 ± 0.03 c	6.1 ± 0.31 a	4.6 ± 0.18 b
Palmitic acid (C16:0)	166.1 ± 3.32 b	215.2 ± 5.21 a	208.7 ± 4.28 a
Stearic acid (C18:0)	58.6 ± 2.34 b	101.4 ± 3.36 a	90.7 ± 3.88 a
Arachidic acid (C20:0)	13.6 ± 0.68 a	13.1 ± 0.59 a	11.9 ± 0.32 ab
Behenic acid (C22:0)	14.1 ± 0.57 ab	16.1 ± 0.69 a	12.7 ± 0.41 b
Lignoceric acid (C24:0)	6.5 ± 0.21 a	6.2 ± 0.24 a	5.3 ± 0.19 b
Total	260.1	358.1	333.9
Unsaturated fatty acids			
Oleic acid (C18:1 n9 c)	35.8 ± 1.07 ab	44.9 ± 2.11 a	34.0 ± 1.55 ab
Linoleic acid (C18:2 n6 c)	331.1 ± 13.24 a	337.6 ± 10.43 a	329.1 ± 11.77 a
α-Linolenic acid (C18:3 n3)	28.3 ± 0.85 a	29.7 ± 0.91 a	30.7 ± 0.90 a
Eicosenic acid (C20:1)	2.6 ± 0.10 a	2.2 ± 0.09 ab	2.1 ± 0.08 ab
Eicosadienoic acid (C20:2)	4.8 ± 0.19 a	4.7 ± 0.17 a	4.6 ± 0.13 a
Eicosatrienoic acid (C20:3 n3)	3.8 ± 0.11 a	nd c	nd
Arachidonic acid (C20:4 n6)	5.1 ± 0.15 b	7.7 ± 0.22 a	7.4 ± 0.20 a
Erucic acid(C22:1 n9)	3.1 ± 0.08 a	3.3 ± 0.06 a	3.1 ± 0.05 a
Docosadienoic acid (C22:2)	nd	nd	1.7 ± 0.02 a
Nervonic acid (C24:1)	3.0 ± 0.06 a	3.2 ± 0.04 a	3.1 ± 0.06 a
Total	417.6	433.1	415.8
Total fatty acids	677.7	791.4	749.7

a—All values are presented as the mean ± SD of pentaplicate determination. Different letters correspond to the significant differences relating to the processing steps using Tukey’s multiple test (*p* < 0.05). b—Processing steps: DMCG, dry mountain-cultivated ginseng; RMCG, red mountain-cultivated ginseng; and FRMCG, fermented red mountain-cultivated ginseng. c—nd: not detected.

**Table 2 molecules-27-04550-t002:** Change of the free amino acid contents during food processing steps of mountain-cultivated ginseng by the cocktail lactic acid bacteria.

Contents a (mg/100 g d.w.)	Processing Steps b		
	DMCG	RMCG	FRMCG
Non-essential amino acids			
Phosphoethanolamine	51.26 ± 2.34 a	43.58 ± 1.12 b	nd c
Proline	53.15 ± 1.47 a	nd	5.64 ± 0.18 b
Hydroxyproline	0.93 ± 0.00 b	6.62 ± 0.32 a	nd
Aspartic acid	58.81 ± 3.33 ab	66.01 ± 2.99 a	59.51 ± 3.02 ab
Serine	24.71 ± 1.57 a	14.74 ± 1.22 b	9.09 ± 1.01 c
Aspartic acid–NH_2_	82.16 ± 2.46 a	37.51 ± 1.10 b	27.81 ± 1.33 b
Glutamic acid	89.01 ± 3.58 a	36.58 ± 1.22 b	27.15 ± 0.99 b
Sarcosine	4.44 ± 0.13 a	2.92 ± 0.12 b	nd
Aminoadipic acid	1.72 ± 0.09 a	0.63 ± 0.03 b	0.71 ± ±0.04 b
Glycine	13.20 ± 0.66 a	4.02 ± 0.11 b	4.91 ± 0.15 b
Alanine	84.61 ± 3.38 a	17.96 ± 0.54 b	11.80 ± 0.47 c
Citrulline	15.01 ± 0.45 a	2.47 ± 0.06 c	5.44 ± 0.12 b
α-aminobutyric acid	12.63 ± 0.33 c	19.67 ± 0.57 b	23.96 ± 1.62 a
Tyrosine	27.80 ± 1.11 a	8.57 ± 0.26 c	9.92 ± 0.30 bc
β-alanine	15.15 ± 0.62 a	12.32 ± 0.33 b	13.54 ± 0.38 b
β-aminoisobutyric acid	9.42 ± 0.38 a	8.97 ± 0.30 ab	8.61 ± 0.27 b
γ-aminobutyric acid	188.99 ± 9.44 a	9.50 ± 0.41 c	100.61 ± 5.03 b
Aminoethanol	11.90 ± 0.47 a	9.47 ± 0.28 a	2.73 ± 0.05 b
Hydroxylysine	29.16 ± 1.46 a	11.95 ± 0.36 b	1.27 ± 0.03 c
Ornithine	16.67 ± 0.67 a	7.21 ± 0.30 b	6.57 ± 0.22 b
3-methylhistidine	nd	0.80 ± 0.00 b	1.22 ± 0.01 a
Arginine	1115.93 ± 44.64 a	440.74 ± 13.22 b	351.31 ± 14.05 c
Total	1,906.66	762.24	671.80
Essential amino acids			
Threonine	29.56 ± 0.88 a	14.36 ± 0.44 b	7.73 ± 0.25 c
Valine	44.39 ± 2.21 a	20.80 ± 1.01 b	22.02 ± 1.11 b
Methionine	13.70 ± 0.41 a	4.40 ± 0.22 b	3.20 ± 0.16 c
Isoleucine	26.90 ± 0.81 a	18.81 ± 0.94 b	23.75 ± 0.95 ab
Leucine	44.89 ± 2.25 a	7.66 ± 0.23 b	7.32 ± 0.20 b
Phenylalanine	27.20 ± 0.77 a	9.50 ± 0.29 b	10.20 ± 0.43 b
Lysine	50.84 ± 1.27 a	8.11 ± 0.24 b	5.90 ± 0.12 c
Histidine	28.54 ± 0.82 a	6.95 ± 0.22 b	4.46 ± 0.15 c
Total	266.02	90.59	84.58
Total amino acids	2,172.68	852.83	756.38
Ammonia	29.16 ± 0.87 a	11.95 ± 0.35 b	11.81 ± 0.47 b

a—All values are presented as the mean ± SD of pentaplicate determination. Different letters correspond to the significant differences relating to the processing steps using Tukey’s multiple test (*p* < 0.05). b—Processing steps: DMCG, dry mountain-cultivated ginseng; RMCG, red mountain-cultivated ginseng; and FRMCG, fermented red mountain-cultivated ginseng. c—nd: not detected.

**Table 3 molecules-27-04550-t003:** Change of the mineral contents during food processing steps of mountain-cultivated ginseng with the cocktail lactic acid bacteria.

Contents a (mg/100 g d.w.)	Processing Steps b		
	DMCG	RMCG	FRMCG
Phosphorus (P)	3.06 ± 0.15 a	2.74 ± 0.13 b	2.88 ± 0.15 ab
Sulfur (S)	2.43 ± 0.10 b	2.53 ± 0.11 ab	2.60 ± 0.09 a
Potassium (K)	16.07 ± 0.64 a	15.97 ± 0.55 a	15.93 ± 0.43 a
Calcium (Ca)	2.66 ± 0.08	2.31 ± 0.04 b	2.39 ± 0.03 ab
Magnesium (Mg)	2.47 ± 0.06 b	2.64 ± 0.08 a	2.63 ± 0.06 a
Iron (Fe)	0.15 ± 0.04 b	0.18 ± 0.05 a	0.17 ± 0.03 ab
Copper (Cu)	0.01 ± 0.00 b	0.02 ± 0.00 a	0.01 ± 0.00 b
Zinc (Zn)	0.07 ± 0.02 a	0.08 ± 0.01 a	0.08 ± 0.01 a
Manganese (Mn)	0.05 ± 0.02 a	0.06 ± 0.00 a	0.06 ± 0.01 a
Natrium (Na)	0.92 ± 0.04 b	0.88 ± 0.05 b	1.30 ± 0.05 a
Aluminium (Al)	0.34 ± 0.01 b	0.41 ± 0.02 a	0.42 ± 0.02 a
Total	28.23	27.82	28.47

a—All values are presented as the mean ± SD of pentaplicate determination. Different letters correspond to the significant differences relating to the processing steps using Tukey’s multiple test (*p* < 0.05). b—Processing steps: DMCG, dry mountain-cultivated ginseng; RMCG, red mountain-cultivated ginseng; and FRMCG, fermented red mountain-cultivated ginseng.

**Table 4 molecules-27-04550-t004:** Change of the ginsenoside contents during food processing steps of mountain-cultivated ginseng by the cocktail lactic acid bacteria.

Contents a (mg/g d.w.)	Processing Steps b		
	DMCG	RMCG	FRMCG
Protopanaxatriol types			
Ginsenoside Rg1 (1)	2.16 ± 0.06 a	1.64 ± 0.08 b	0.52 ± 0.02 c
Ginsenoside Re (2)	4.4 ± 0.20 a	3.25 ± 0.10 b	0.95 ± 0.04 c
Ginsenoside Rf (4)	0.79 ± 0.03 a	0.73 ± 0.02 a	0.62 ± 0.01 b
Ginsenoside F5 (5)	nd c	0.02 ± 0.00 a	nd
Ginsenoside F3 (6)	nd	nd	nd
Ginsenoside Rg2 (7)	0.48 ± 0.01 c	0.73 ± 0.03 b	1.01 ± 0.05 a
Ginsenoside Rh1 (8)	0.96 ± 0.04 b	1.19 ± 0.51 b	1.41 ± 0.48 a
Ginsenoside F1 (11)	nd	nd	nd
Protopanaxatriol (18)	nd	0.64 ± 0.02 b	0.81 ± 0.04 a
Total	8.79	8.2	5.32
Protopanaxadiol types			
Ginsenoside Rb1 (9)	8.7 ± 0.35 a	9.01 ± 0.36 a	4.69 ± 0.15 b
Ginsenoside Rc (10)	4.2 ± 0.17 a	4.31 ± 0.16 a	2.05 ± 0.09 b
Ginsenoside Rb2 (12)	1.62 ± 0.08 a	1.47 ± 0.04 b	0.78 ± 0.02 c
Ginsenoside Rb3 (13)	0.32 ± 0.00 a	nd	nd
Ginsenoside Rd (14)	1.83 ± 0.09 b	2.21 ± 0.11 a	1.43 ± 0.04 c
Ginsenoside Rd2 (15)	1.81 ± 0.08 b	2.35 ± 0.10 ab	2.77 ± 0.13 a
Ginsenoside F2 (16)	0.31 ± 0.01 c	1.02 ± 0.05 b	2.27 ± 0.11 a
Ginsenoside Rg3 (17)	0.36 ± 0.0 2 c	0.77 ± 0.03 b	1.93 ± 0.09 a
Compound K (19)	0.5 ± 0.02 c	1.68 ± 0.05 b	4.13 ± 0.03 a
Ginsenoside Rh2 (20)	nd	0.26 ± 0.00 b	0.51 ± 0.02 a
Protopanaxadiol (21)	0.59 ± 0.02 a	0.31 ± 0.01 b	0.50 ± 0.02 a
Total	20.24	23.39	21.06
Oleanane types			
Ginsenoside Ro (3)	2.22 ± 0.07 a	0.77 ± 0.02 c	1.09 ± 0.04 b
Total	2.22	0.77	1.09
Total ginsenosides	31.25	32.36	27.47

a—All values are presented as the mean ± SD of pentaplicate determination. Different letters correspond to the significant differences relating to the processing steps using Tukey’s multiple test (*p* < 0.05). b—Processing steps: DMCG, dry mountain-cultivated ginseng; RMCG, red mountain-cultivated ginseng; and FRMCG, fermented red mountain-cultivated ginseng. c—nd: not detected.

**Table 5 molecules-27-04550-t005:** Change of the volatile flavor compounds during food processing steps of mountain-cultivated ginseng by the cocktail lactic acid bacteria.

Volatile Flavor Compounds a (%)	Processing Steps b			Odor Description
	DMCG	RMCG	FRMCG	
α-pinene	0.15 ± 0.02 a	nd c	nd	Cool, Ginseng
β-myrcene	0.03 ± 0.00 a	nd	nd	
1-piperidinecarboxaldehyde	0.02 ± 0.00 a	nd	nd	
Limonene	0.04 ± 0.01 a	nd	nd	Mint
2-methoxy-3-1-methyl ethyl pyrazine	0.02 ± 0.00 a	nd	nd	
2,3,5,6-tetramethyl pyrazine	0.02 ± 0.00 a	nd	nd	
Maltol	nd	0.07 ± 0.01 b	0.10 ± 0.02 a	Sweat
Bornyl acetate	0.03 ± 0.00 a	nd	nd	
Bicycloelemene	0.11 ± 0.02 b	0.84 ± 0.04 a	0.92 ± 0.03 a	
1-methyl ethyl benzene	3.97 ± 0.13 a	2.76 ± 0.08 b	nd	
Cedrene	nd	nd	0.17 ± 0.02 a	
Selina-4,11-diene	9.35 ± 0.28 a	nd	nd	
Drima-8,12-9,11-diene	3.42 ± 0.11 b	2.54 ± 0.08 c	4.08 ± 0.20 a	
β-panasinsene	17.28 ± 0.69 c	22.69 ± 0.91 b	31.61 ± 0.95 a	Ginseng
Berkheyaradulene	nd	nd	4.91 ± 0.22 a	
β-elemene	13.60 ± 0.54 a	8.82 ± 0.44 b	13.70 ± 0.66 a	Hurb
4-1-methyl ethyl benzaldehyde	1.90 ± 0.07 a	1.31 ± 0.06 b	2.11 ± 0.11 a	Sweat
α-gurjunene	1.99 ± 0.08 a	1.11 ± 0.05 b	1.88 ± 0.07 a	
δ-selinene	nd	1.71 ± 0.08 a	nd	
Valencene	nd	nd	1.35 ± 0.07 a	
Calarene	12.94 ± 0.63 a	9.54 ± 0.19 a	nd	
Aromadendrene	nd	1.12 ± 0.05 b	2.39 ± 0.12 a	
Selina-5,11-diene	nd	nd	0.80 ± 0.03 a	
β-gurjunen	nd	16.47 ± 0.49 a	nd	
Humulene	nd	7.28 ± 0.29 a	nd	
Alloaromadendrene	1.64 ± 0.05 b	1.39 ± 0.06 c	2.60 ± 0.13 a	
Caryophyllene	12.90 ± 0.64 ab	8.50 ± 0.26 b	13.82 ± 0.59 a	
β-selinene	3.77 ± 0.11 a	3.43 ± 0.10 ab	3.53 ± 0.14 a	
β-neoclovene	3.17 ± 0.12 a	1.95 ± 0.09 b	nd	
Germacrene	0.22 ± 0.01 a	0.12 ± 0.00 b	0.13 ± 0.01 b	
α-selinene	nd	2.85 ± 0.11 b	9.36 ± 0.28 a	
Bicyclogermacrene	11.97 ± 0.39 b	4.90 ± 0.24 a	4.81 ± 0.22 a	
Butylatedhydroxy toluene	0.50 ± 0.02 b	nd	0.75 ± 0.03 a	
α-amorphene	0.09 ± 0.01 a	nd	nd	
δ-cadinene	0.39 ± 0.01 c	0.50 ± 0.02 b	0.94 ± 0.04 a	
β-guaiene	nd	nd	0.02 ± 0.00 a	
1,2,4,4-tetramethyl cyclopentene	0.13 ± 0.02 a	nd	nd	
Cubedol	0.13 ± 0.02 a	nd	nd	
Nerolidol	0.21 ± 0.03 a	0.10 ± 0.02 b	nd	

a—All values are presented as the mean ± SD of pentaplicate determination. Different letters correspond to the significant differences relating to the processing steps using Tukey’s multiple test (*p* < 0.05). b—Processing steps: DMCG, dry mountain-cultivated ginseng; RMCG, red mountain-cultivated ginseng; and FRMCG, fermented red mountain-cultivated ginseng. c—nd: not detected.

## Data Availability

The data that support the findings of this study are available from the corresponding author, upon reasonable request.

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
