# Peer review of "Changes in Chemical Compositions and Antioxidant Activities from Fresh to Fermented Red Mountain-Cultivated Ginseng"

_molecules, 2022, doi:10.3390/molecules27144550_

Round 1

Reviewer 1 Report

The manuscript: Changes in chemical composition and antioxidant activity from fresh to fermented red mountain-cultivated ginseng seems to lack innovation, does not provide the authors' thoughts and comments, and is more like mere data compilation. The specific comments are as follows for the authors' reference.

Introduction: authors should emphasize the objectives of the research and better describe the role of the compounds investigated in the paper, as well as the biological activities of the ginseng.

Material and methods: Too many authors self-citations for every method. Please correct it. Cite the original paper for the methods that you used.  

Conclusion: authors should emphasize the novelty and significance of their work. Also the future perspectives of the ginseng application in food industry should be added in manuscript.

Author Response

[1] The manuscript: Changes in chemical compositions and antioxidant activities from fresh to fermented red mountain-cultivated ginseng seems to lack innovation, does not provide the author’s thoughts and comments, and is more like mere data compilation. The specific comments are as follows for the author’s reference.

First of all, we are grateful reviewer’s advices in reviewing our manuscript. Also, we thank the reviewer for giving us the opportunity to revise our manuscript. I heartily subscribe to the opinion of reviewer. According to the reviewer’s suggestion, the corrected parts in the revised manuscript are colored with red.

Moreover, we rewrite the many sentences to help understanding (the author’s thoughts and comments) in the revised manuscript. The corrected or revised sentences were reconstituted as following:

â–º (Example) (A) (P 3 line 98, 105-108) 2.1. Changes in physicochemical properties part: Based on the above results, we confirmed that the environmental stresses of processing methods may be important influences in variation ratios regarding action glycoside hydrolases from DMCG [20,21]. (B) (P 5 line 145-147) 2.2. Changes in nutrients part: --- and their differences may be considered as the different kinds of ginseng and microorganisms as well as fermentation environmental conditions. (P 7 line 178-180) 2.2. Changes in nutrients part: The fluctuations of individual and total AAs may be also associated with various parameters concern to the process techniques [31]. (P 8 line 194-198) 2.2. Changes in nutrients part: In particular, we believe that the increase rates of minerals in the FRMCG sample may be related to the degradation and transformation of other nutritional compositions through microbal growth during fermentation process [9,10,32]. (P 8 line 217-221) 2.3. Changes in TP and TF contents part: These phenomena may be positively correlated with changes in acid or estrolytic enzymes through the release of insoluble ester bounds (including bound phenolic and/or flavonoids) [11,35].

â–º Additionally, we rewrite the many sentences to help understanding. The revised parts are colored with red.

[2] Introduction: authors should emphasize the objectives of the research and better describe the role of the compounds investigated in the paper, as well as the biological activities of the ginseng.

I heartily subscribe to the opinion of reviewer #1. Although many researchers reported that the bioconversion of ginsenosides in ginseng, there are studied a little that comprehensive fluctuations of nutritional metabolites, antioxidant activities, physicochemical properties, and influential factors on antioxidant from various experimental methods through the processing steps of ginseng sprouts. We drew up detailed sentences regarding the originality and novelty of this research in “Introduction” section. We have tried to revise the manuscript as much as possible in line with suggestions made by the reviewer#1. The corrected or new parts were colored with red in revised manuscript. I hope the improved version will be acceptable for publication in molecules.

â–º (Added sentences in introduction) introduction correction to clear objective of this study: (P 3 line 72-75) Even though there have been several articles on gensenoside and VFC contents with environmental conditions, to the best of our knowledge, there have been no comprehensive information MCG using processing techniques. (P 3 line 84-87) For these above observations, our work was designed to demonstrate the primary and secondary metabolites as well as antioxidant properties under processing skills of aging and fermentation from MCG.

[3] Material and methods: Too many authors self-citations for every method. Please correct it. Cite the original paper for the methods that you used.

I heartily subscribe to the opinion of reviewer. According to the reviewer’s opinion, we have corrected to the appropriate sentences in “Material and methods” section regarding the originality and novelty on the important factors of our results.

â–º (Corrected references: Example) (1) (P 16 line 333) 3.3 Determination of physicochemical properties: Lee et al. [20] → Hua et al.and Lee et al [5,22] (2) (P 16 line 425) 3.4 Analysis of FAs: ---previously reported data [1,21] → ---previously reported data [23,26] (3) (P 17 line 439) 3.5 Analysis of FAAs: ---according to Hwang et al. [32] → ---according to Wan et al. [8] and Kuo et al. [31]. (4) (P 17 line 449-500) 3.6 Analysis of Minerals: ---according to Lee et al. [1] → ---according to Park et al. [32]. (5) (P 17 line 468) 3.7 Analysis of TP and TF contents and MR products: ---as described previously [1] → --- as described previously [44]. (6) (P 17 line 481) 3.8 Analysis of ginsenosides: ---as previously described in Lee et al. [1] → ---as previously described in Shibata et al. [45]. (7) (P 18 line 491-492) 3.9 Analysis of VFCs: ---according to Cho et al. [42] → --- according to El-Aty et al. [14] and Cho et al [41]. (8) (P 18 line 514-515) 3.10 Determination of antioxidant effects: ---according to Lee et al. [21] and Lee et al. [33] → --- according to Cho et al. [29] and Lee et al [9].

[4] Conclusion: authors should emphasize the novelty and significance of their work. Also the future perspectives of the ginseng application in food industry should be added in manuscript.

According to the reviewer’s advice, we have added to the appropriate sentences in “Conclusion”parts in the revised manuscript to improve the originality of manuscript. The added parts in the revised manuscript are colored with red.

â–º (Added sentence: Example) (1) (P 18 line 528-531) Our work demonstrated for the first time that an important information concern to changes in primary (FFAs, FAs, and minerals) and secondary (ginsenosides and VFCs) metabolites as well as antioxidant properties of processed MCG by aging and fermentation. (2) (P 19 line 555-559) Also, we believed that processed DMCG sources can be key materials in the ginseng quality for the development of human functional foods. Future researches are needed to document the beneficial human health biological activities in processed DMCG for advancement of food and medical industrious.

Reviewer 2 Report

The work is well described, the experimental design clear and the results accurately discussed.

I have one guestion. Is it possible to determine antioxidant activity by other methods?

Author Response

[1] The work is well described, the experimental design clear and the results accurately discussed.

First of all, we are grateful reviewer’s advices in reviewing our manuscript. Also, we thank the reviewer for giving us the opportunity to revise our manuscript. We rewrite (or delete) the many sentences and appropriate scientific literatures in each section of the revised manuscript to help understanding. The revised parts are colored with red. I hope the improved version will be acceptable for publication in molecules.

[2] I have one question. Is it possible to determine antioxidant activity by other methods?

I heartily subscribe to the reviewer’s opinion regarding this above sentence in manuscript. Generally, it is well established that the phytochemicals and their extracts of natural plants and food sources have been considered to be significant antioxidants. Due to the simple, convenient and fast features of in vitro antioxidant activity assay, a variety of in vitro methods have been developed and widely used to determine the antioxidant activities of phytochemicals and natural extracts in the literatures, such as DPPH radical-scavenging activity, oxygen radical scavenging capacity (ORAC), total radical-trapping antioxidant parameter (TRAP), peroxyl radical scavenging capacity (PSC) assay, etc. In particular, the hydroxyl, ABTS and DPPH radical scavenging abilities and FRAP method are commonly used to demonstrate the antioxidant properties because of their simple reproducibility and quality control. Therefore, we evaluated changes in antioxidant properties against hydroxyl, ABTS and DPPH radicals, FRAP assay. Future work is needed to examine the bioactive materials and their potential human beneficial abilities including various antioxidant methods from processed DMCG sources for the development of functional foods. We will do more research of in vivo and in vitro antioxidant properties, based on chemical assay and culture model.

Reviewer 3 Report

The paper by Lee et al. studied the changes in chemical composition in korean red ginseng during a fermentation process. It addresses a very timely and interesting topic, because Korean ginseng, a well known powerful medicinal herb, has been shown to increase its pharmacological properties through transformation into red and black ginseng. Additionally the bioactivities can be modified and through fermentation techniques as shown in this paper. It is definitely interesting for the readership of this journal and should be accepted for publication, after some minor revisions:

- Please check the English language in general.  

- Line 78-79: The sentence is incomplete. "processing steps of" what?

- Line 95-96:  The sentence needs to be checked. Word order is wrong. 

- names of compounds have to be corrected: Table 2: Phosphoenthanolamine, an h is missing. In the whole Paper: Panaxatriol, an a is missing.

- Line 183 to 185: The meaning of this sentence is not clear. If the authors found that the mineral content increases in FRMCG, then why they suggest that it is transformed into other nutrients? Would that not mean, that the mineral content is lower in FRMCG, because it is transformed into other nutrients by the microorganisms, at the time you do the analysis? Please clarify.

- In line 231 and 232: the meaning is not clear. I guess the word "increased" is missing?

"Rg2 (0.48 → 0.73 → 1.01), Rh1 (0.96 → 1.19 → 1.41 mg/g), and 

232 PPT (nd → 0.64 → 0.81 mg/g), increased respectively"

Author Response

The paper by Lee et al. studied the changes in chemical composition in korean red ginseng during a fermentation process. It addresses a very timely and interesting topic, because Korean ginseng, a well-known powerful medicinal herb, has been shown to increase its pharmacological properties through transformation into red and black ginseng. Additionally the bioactivities can be modified and through fermentation techniques as shown in this paper. It is definitely interesting for the readership of this journal and should be accepted for publication, after some minor revisions:

First of all, we are grateful reviewer’s advices in reviewing our manuscript. Also, we thank the reviewer for giving us the opportunity to revise our manuscript. I heartily subscribe to the opinion of reviewer. According to the reviewer’s suggestion, the corrected parts in the revised manuscript are colored with red.

[1] Please check the English language in general.

⇒ The written English in revised manuscript was examined carefully by English native. Also, we proceeded with the English correction in enago.

[2] Line 78-79: The sentence is incomplete. "processing steps of" what?

⇒ I apologize for having careless mistake while writing. According to the reviewer’s suggestion, we rewrite this sentence in the revised manuscript to help understanding.

â–º (Corrected sentence) (P 3 line 84) ---according to MCG processing steps of → --- according to MCG processing steps.

[3] Line 95-96: The sentence needs to be checked. Word order is wrong.

⇒ I apologize for having careless mistake while writing. According to the reviewer’s suggestion, we rewrite this sentence in the revised manuscript to help understanding.

â–º (Corrected sentence) (P 3 line 103-105) “In addition, the contents of soluble solids a little increased to RMCG sample (3.2 brix) and after decreased slightly in FRMCG (Figure 2)” → “In addition, the contents of soluble solids increased a little in the RMCG sample (3.2 brix) and decreased slightly in FRMCG (Figure 2)”

[4] Names of compounds have to be corrected: Table 2: Phosphoenthanolamine, an h is missing. In the whole Paper: Panaxatriol, an a is missing.

⇒ I apologize for having careless mistake while writing. According to the reviewer’s suggestion, we rewrite this sentence in the revised manuscript (including Table 2) to help understanding.

â–º In Table 2, “Phosphoentanolamine” has been corrected to “Phosphoenthanolamine”. In the whole paper, Panaxtriol was change Panaxatriol (Table 4, Figure 5). In addition, Panaxdiol was modified to Panaxadiol (Line 253, Table 4).

[5] Line 183 to 185: The meaning of this sentence is not clear. If the authors found that the mineral content increases in FRMCG, then why they suggest that it is transformed into other nutrients? Would that not mean, that the mineral content is lower in FRMCG, because it is transformed into other nutrients by the microorganisms, at the time you do the analysis? Please clarify.

⇒ According to the reviewer’s suggestion, we rewrite this sentence in the revised manuscript to help understanding.

â–º (Corrected sentence) (P 8 line 194-198) “We believe that the increase in minerals in the FRMCG step may be converted to other nutrition constituents through microbial growth during fermentation” → “In particular, we believe that the increase rates of minerals in the FRMCG sample may be related to the degradation and transformation of other nutritional compositions through microbal growth during fermentation process”

[6] In line 231 and 232: the meaning is not clear. I guess the word "increased" is missing? "Rg2 (0.48 → 0.73 → 1.01), Rh1 (0.96 → 1.19 → 1.41 mg/g), and 232 PPT (nd → 0.64 → 0.81 mg/g), increased respectively"

⇒ I apologize for having careless mistake while writing. According to the reviewer’s suggestion, we rewrite this sentence in the revised manuscript. The corrected parts in the revised manuscript are colored with red.

â–º (Corrected sentence) (P 9 line 250-252 " while the ginsenoside Rg2, Rh1, and PPT contents increased with 0.48 → 0.73 → 1.01, 0.96 → 1.19 → 1.41, and nd → 0.64 → 0.81 mg/g, respectively.”

Round 2

Reviewer 1 Report

/

Author Response

Response to reviewers

Changes in chemical compositions and antioxidant activities from fresh to fermented red mountain-cultivated ginseng

Dear Meta Miao, Ph.D

Editor
Molecules

Reviewer’s comments and suggestions for authors

Reviewer # 1

The authors have included the modifications suggested by the reviewers, but in order to be published, the following changes must be made: Page 8, line 209: L. plantarum in italics Page 11, line 285: it is Leuconostoc (not Leuconsotoc) Table 2: Hydroxyproline (not Hydroxyprline) Table 2: 3-methylhistidine (not 3-Methylhistidine) Table 3: potassium, not kalium Table 3: Please, to delete "c nd: not detected." in the footnote. Table 5 and in the manuscript: All volatile compound' names after a number and a hyphen must be written in lowercase letters (e.g.: α-pinene, β-myrcene, 1-piperidinecarboxaldehyde, 2,3,5,6-tetramethyl pyrazine, etc.) Page 18, line 523: "...FeCl3 at 10:1:1 (v/v/v)". Please, to add: "(v/v/v)" Page 18: Pentaplicate determinations are commendable, but modifications and evolutions must be ratified by a statistical analysis to validate or reject whether these different data really correspond to an effective change. A statistical analysis (Student, Fisher, ANOVA, etc.) must be included. Consequently, this analysis should be included in the tables and figures with superscript letters and in the text, when commenting on an increase or decrease, indicate whether they are statistically different, for example, for a 95% probability.

  • Frist of all, we thank you for giving us the opportunity to resubmit. According to the reviewer’s opinion, we adapted in the revised manuscript. Corrected sentences were marked up using the “track changes” in the manuscript. And the corrects are summarized as follows.
  • Page 8, line 209: L. plantarum in italics plantarum (Page 8, line 217)
  • Page 11, line 285: it is Leuconostoc (not Leuconsotoc)Leuconostoc (Page 6, line 297)
  • Table 2: Hydroxyproline (not Hydroxyprline) → Hydroxyproline (Page 6, Table 2)
  • Table 2: 3-methylhistidine (not 3-Methylhistidine) → 3-methylhistidine (Page 6, Table 2)
  • Table 3: potassium, not kalium → Potassium (Page 7, Table 3)
  • Table 3: Please, to delete "c nd: not detected." in the footnote. → “cnd: not detected” part was deleted. (Page 8, Table 3)
  • Table 5 and in the manuscript: All volatile compound' names after a number and a hyphen must be written in lowercase letters (e.g.: α-pinene, β-myrcene, 1-piperidinecarboxaldehyde, 2,3,5,6-tetramethyl pyrazine, etc.)

→ “α-pinene, β-myrcene, 1-piperidinecarboxaldehyde, 2-methoxy-3-1-methyl ethyl pyrazine, 2,3,5,6-tetramethyl pyrazine, 1-methyl ethyl benzene, β-panasinsene, β-elemene, 4-1-Mmethyl ethyl benzaldehyde, α-gurjunene, δ-selinene, β-gurjunen, β-selinene, β-neoclovene, α-selinene, α-amorphene, δ-cadinene, β-guaiene, and 1,2,4,4-tetramethyl cyclopentene” (Page 13, Table 5)

  • Page 18, line 523: "...FeCl3 at 10:1:1 (v/v/v)". Please, to add: "(v/v/v)"

→ “(v/v/v)” is inserted into the manuscript. (Page 18, line 544)

  • Page 18: Pentaplicate determinations are commendable, but modifications and evolutions must be ratified by a statistical analysis to validate or reject whether these different data really correspond to an effective change. A statistical analysis (Student, Fisher, ANOVA, etc.) must be included. Consequently, this analysis should be included in the tables and figures with superscript letters and in the text, when commenting on an increase or decrease, indicate whether they are statistically different, for example, for a 95% probability.

→ According to the reviewer’s opinion, we performed statistical processing. And, it adapted the data in the revised manuscript.

  • (Added sentence)

→ “Different letters correspond to the significant differences relating to the processing steps using Tukey's multiple test (p < 0.05).” (Page 4-5, Figure 2, line 125-127; Page 6, Table 1; Page 7, Table 2; Page 8, Table 3; Page 8, Figure 3, line 224-225; Page 11, Table 4; Page 13, Table 5; Page 15, Figure 6, line 379-381)

→ “3.11. Statistical analysis

The nutrient compounds and antioxidant activities were expressed as the mean ± SD (standard derivation) of five measurements. The significant differences among samples were determined by Tukey's multiple test (p < 0.05) using the Statistical Analysis System (SAS) software (ver. 9.4; SAS institute, Cary, NC, USA).” (Page 18-19, line 549-554)
